# The Digital Divide Is Aging: An Intergenerational Investigation of Social Media Engagement in China

**DOI:** 10.3390/ijerph191912965

**Published:** 2022-10-10

**Authors:** Yuqiong Zhou, Tao He, Feng Lin

**Affiliations:** 1School of Media and Communication, Shenzhen University, Shenzhen 518060, China; 2Depart of Educational Technology, College of Education, Shenzhen University, Shenzhen 518060, China

**Keywords:** digital divide, grey digital divide, older adults, generational differences, social media, WeChat

## Abstract

The aging-based digital divide has gained increased research attention in recent years. This research focused on two important aspects of the aging society, the digital divide and the intergenerational differences, with regard to social media usage in China. The data were collected through a household interview, using a nationally representative sample (n= 3790) from 58 cities in mainland China. The present study investigated the association between differentiated social media usage with demographic characteristics, social economic status, physical and mental health, and social media perceptions. The results show a significant digital divide and generational differences. The existing socio-economic inequalities and demographic variance largely explained older adults’ social media access and diversity of social media engagement. Social media perceptions play more essential roles in the intensity of social media engagement for each generation group. This study contributes a better understanding of the manner in which different factors are associated with different generations’ social media engagement. A discussion is included regarding the necessity to close the aging-based digital gap in order to cope with the issue of rapid aging.

## 1. Introduction

Digital technologies are increasingly evolving while the population is aging rapidly, especially in China. In recent years, China has aged more dramatically than almost any other country, and the older population aged 60 and over will peak at about 430 million by 2050, representing a quarter of the total population, and meaning China will have the oldest population in the world [1]. Meanwhile, the new digital media-saturated environments not only provide opportunities but also challenge older adults’ later life [2].

Recent studies have emerged to address the inequalities among generational cohorts with special attention in younger adults in comparison with older adults, and this digital gap is also called the “grey digital divide” [3,4]. However, prior studies on this specific divide have mainly focused on the level of Internet access in many developed countries. For instance, among young people, the Internet access rate has reached 95% in the United States [5], and more than 97% in western European countries such as Germany and Switzerland [3].

In contrast, among older people aged 65 and over, Internet access is practically saturated at less than 57% in the United States and Switzerland [3,6]. Compared to an even lower Internet access rate of older adults in China, the grey digital divide is still an issue that needs to be examined. The diffusion of new media is neither a self-starting nor a random process. Instead, Internet and social media access is strongly associated with various socio-demographic variables such as education, economic, gender, and age [3,7,8].

As the mobile Internet and social media services are developing, the major trend of this increase in online services is “all in one”, which means all Web 2.0 functions and services are embedded in one form of social media, such as WeChat in China [9]. Compared with traditional social media such as Facebook, WeChat has merged even more functions and services into one application. WeChat has gained significant popularity in China and the majority of Chinese Internet users access and engage it continually in everyday life. It has been reported that WeChat has obtained more than one billion active users per month [10].

Although a considerable amount of literature has shown that the use of social media by the younger generation would lead to a series of social consequences, such as increasing civic involvement and maintaining of social relationships, research into seniors’ social media usage is still rather limited [8, 11]. Since social media have relevance for economic, cultural, and private lives, the disparity of their use leads to inequalities at the level of individuals related to generational groups. This paper presents a cross-generational approach to fully investigate the patterns of the intergenerational digital divide, especially in relation to social media engagement.

Understanding generational usage differences and their determinants are critical for identifying larger patterns of new media adoption and the digital divide, and would ensure that any benefits associated with their use are equally available to all members of society. In order to better understand the ramifications of social media for the everyday experiences of three generational groups, this study addressed four important research questions as follows:

Q1: Is there a digital divide in terms of social media access across three generational cohorts?

Q2: What factors explain older adult’s social media access if the grey digital divide exists?

Q3: What factors are associated with the diversity of social media engagement and how do these factors vary by generation?

Q4: What factors are associated with the intensity of social media engagement and how do these factors vary by generation?

### 1.1. Theoretical Perspectives and Related Works

#### 1.1.1. The Multifaceted Nature of the Digital Divide

The digital divide, as a metaphor, was originally coined to identify the inequalities between the technological haves and have-nots [12]. These inequalities separate individuals, households, and geographic areas at different socio-economic levels, into those individuals who are able to take advantage of the ICT opportunities for a wide range of activities and those who are not [6,7]. Because of sustained attention in mass media, the digital divide has frequently been a part of the agenda of social, political, and scholarly discussions.

A substantial body of the early literature on the digital divide studied the inequities in the access to the Internet and access to social media, which have been called the first and second levels of the digital divide [4,13]. Recently, the term “third digital divide” has drawn growing attention in both academia and industry, since specific online activity such as social media engagement was under-researched [14]. Even though engaging with social media brings social benefits, such as helping individuals to create, maintain, and promote social capital, the differences in the use of these social media by different age groups have not been studied in sufficient detail [15].

Prior studies have found that individuals may be involuntarily excluded from using ICT due to a lack of opportunities, such as socio-economic status [16]. This disadvantaged socio-economic status greatly contributes to the first-level digital divide, namely the inequalities of Internet access [17]. However, when individuals are online, the usage pattern shifts from “haves and have-nots” to “want and want-not”, who would more voluntarily engage specific new media activity. Thus, the second and third level digital divides are shaped not only by socio-economic background, but also by personal perceptions and needs [14]. However, limited empirical research is available to provide an up-to-date look at the third digital divide related to social media usage.

#### 1.1.2. Generational Cohorts

Generations are represented by age groups. Generational cohort theory is based on the basic assumption that the societal, political, and economic circumstances in which a generation grows up will continue to influence a generation’s behavior later in life to a large extent [18]. Each generational cohort refers to individuals who are born during a specific period of time, and share similar experiences, attitudes, and values, which will remain relatively stable during an individual’ life course [19].

The most frequently used generational cohorts in the literature are baby boomers, generation X, and generation Y [20]. These generational cohorts were coined to classify specific age groups; for instance, baby boomers refer to the generation cohort born between 1946 and 1964, when the post-war economy was thriving, which led to a dramatic population increase [21]. Accordingly, the Chinese generational cohorts can be also classified into three cohorts based on two distinct historical events that occurred in China, i.e., the Cultural Revolution in the 1960s and the Economic Reforms since 1978. The first generation cohort is the Pre-Cultural Revolution Generation, which is used to describe the members who were born before the Cultural Revolution and entered the formative years of tremendous social unrest. The second generation cohort is the Post-Cultural Revolution Generation, which refers to the members born after the beginning of the Cultural Revolution. This group of people grew up during the beginning of economic reform and takes pride in personal accomplishment and self-interest [22].

The third generation is the Post-Reform and Open generation. Members of this generational cohort became teenagers during China’s economic reform years and experienced the dramatic economic growth and social transition [23]. They enjoy individual opportunities and are well-exposed to Western popular culture and digital media programs. Additionally, these generational cohorts echo the global discourses of “digital native”, “digital immigrant”, and “digital refugee”, which describe the different generations according to the time at which they entered the digitalized world. Younger people, middle-aged adults, and the grey population can be generally classified into these three categories according to digital globalization [24].

Prior aging studies have determined different bottom lines for the definition of the “older generation” or “older citizens”. Many researchers defined people aged over 50 as senior [25], whereas others classified people aged 55 and over as the older generation [26], or defined older adults as those who are aged 60 and over [27]. Considering that the retirement age for Chinese people averages 55, and most Chinese gerontology researchers define Chinese seniors as aged 55 and over [28], the present study followed this approach and defined older adults as those aged 55 and over.

According to the generational cohort theory and aging studies, it is very important to gain an additional understanding of the digital media users, since their motives, attitudes, and behaviors may be shaped by a cohort period of time having certain societal, economic, and even media environments [29]. Media usage motives and behaviors tend to reveal a homogeneity within each generational cohort and a heterogeneity across generational cohorts [29]. Therefore, employing these approaches, the present study classified the full sample into three generational groups by their ages according to the Chinese social transition, aging status, and digitalization: the older generation (aged 55 and over), middle-aged generation (aged 40–54), and younger generation (aged 14–39).

#### 1.1.3. Aging and New Media Usage

China is facing the most urgent aging challenge because it has the fastest aging population in the world [30]. Meanwhile, the current new media environment and digital solutions provide more opportunities to support older adults’ later life [2]. Prior research argues that new media play a growingly important role in active aging. For instance, digital technologies assist independent living for older people with physical disability and mental health impairment [31]. Digital access and social media connect older adults’ daily life with extensive active social participation and social inclusion [32]. Living through open and personalized digital solutions may improve the quality of life for older people [2].

Furthermore, motive patterns among younger generations may not apply to aging populations, because seniors’ life styles and circumstances are distinct from those of younger adults. For example, older adults’ impaired physical and mental health conditions often hinder new media adoption and challenge older adults’ actual usage, even when they have high ownership of digital media [33]. Later-life perceptions, such as perceived social influence among peers, may spur social participation and then lead to possible social media use [11]. These factors are rarely examined in the studies of younger adults, but are essential to include when studying digital inequalities among aging populations.

## 2. Design and Methods

### 2.1. Data Collection and Sample

All data were collected through face-to-face interviews by 60 trained interviewers. Three surveys were specifically designed for different generational groups in the household unit.

The sampling frame followed the prior major national research of the Chinese Longitudinal Aging Study Survey (CLASS) and China Longitudinal Healthy Longevity Survey (CLHLS) to represent the national population according to the population distribution and economic development [34]. Specifically, the national representative samples were collected from 28 provinces and municipalities, which covered 97.8% of the national population according to the China Population Statistics Yearbook 2007, and the national baseline surveys conducted from December 2017 to April 2018.

A total of 58 city-level units were selected using quota sampling with probability proportional to size (PPS), and stratified by region and city gross domestic product (GDP) from east to west in mainland China. Hence, 40 households for each first-tier city and 20 households for every other city were assigned. At the city level, four highly developed first-tier cities (e.g., Beijing, Shanghai), seven moderately developed cities, such as provincial capitals, and 47 less-developed normal cities were selected. Eventually, 1160 households were selected, which generated a total of 3790 valid samples from 956 households (the response rate was 82.4%) and represented three generational cohorts of mainland China.

Since older adults had the highest priority in our surveys (because they are hard to reach), we ensured that all older adults (age 55+) in each household must be interviewed. Then, the investigator interviewed at least one middle-aged adult (age 40–54) and one young person (age 14–39) if applicable. In addition to three-generation families, there were a large number of two-generation families in our sample. For such families, only the older and younger participants were interviewed. Within each household unit, participants aged 55 and over were asked to respond to the older generations’ version of the survey (n = 1399); participants aged 40 to 54 who served as parents were asked to respond to the middle-aged generations’ survey (n = 543); and participants aged from 14 to 39 were asked to respond to the younger generations’ survey (n = 1109).

### 2.2. Measurements

#### 2.2.1. Outcome Variables

First, all participants were asked whether have access to Internet, then the central focus of *Social Media access* was measured by a question about whether one had ever used WeChat (No = 0, Yes = 1).

*Social Media Engagement* involves the individual’s use of 18 functions of WeChat for 3 types of purpose. The first category is social networking, which includes 8 functions (i.e., instant messaging by text or voice, voice call, video call, add friends, accept friend invitation, likes and comments, group chat). The second category is information sharing and creation, which includes 3 functions (i.e., moments posting, browsing and reposting the official account). The third category is payment services, which includes 3 functions (i.e., online transfer, instant pay, red packets). The last category is advanced creation, which include 4 functions (i.e., create official account, applet, search message, find chat records). Each individual was asked a binary question such as “Do you use moments posting” (No = 0, Yes = 1), and a final composite measure of diversity of WeChat activities was created by summing the items, ranging from 0 to 18, with a higher value indicating a greater diversity of engaging social media activities.

Then, we used *WeChat use intensity* scale, which was adapted from the validated Facebook Intensity scale [35], with three dimensions (persistence, self-expression, and overuse) included to measure the intensity of WeChat. The *persistence* scale comprised four items such as “If I could visit only one APP in my smartphone, it would be WeChat” and “I feel bad if I don’t check my WeChat daily”. The *self-expression* scale had three items, for example, “I like posting my WeChat moments” and “It is important for me to update my profile”. The *overuse* scale had four items, for example, “I always spend more time on WeChat than I planned” and “Sometimes I would use WeChat till midnight”. All items utilized a five-point Likert scale ranging from 1 (strongly disagree) to 5 (strongly agree). The average scores were calculated for the final analysis. The scale has good consistency and reliability (Cronbach α= 0.875).

#### 2.2.2. Predictor Variables

*Demographic Characteristics* was documented in module one of the surveys, which gives details of participants’ gender (*fe**male* = 0, *male* = 1), age (year), marital status (*single/widowed without partner* = 0, *cohabiting with partner* = 1, *married and with spouse =* 1), residency (*rural* = 0, *urban* = 1), number of children, and neighborhood, which refers to the distance to the city center using an ordinal scale ranging from 1 to 6 (*in the rural community* = 0, *in the city center community* = 1)

*Social economic status* variables measured education, professional occupation, professional rank, and living region. In terms of education level, seven categories ranged from 1 (*illiterate*) to 5 (*postgraduate and beyond).* A professional ranked question was used to determine respondents’ work position from low to high, ranging from 1 (ordinary worker) to 4 (senior executive). Respondents’ living regions were documented to indicate the level of the region’s economic development according to the GDP index, ranging from 1 (*less developed*) to 3 (*highly developed*).

*Physical and Mental health* test was only included in the aged survey, which followed the national “China Longitudinal Health Longevity Survey”. Physical health was measured by four indicators: self-reported health, vision, hearing, and having a disability. Self-reported health was a subjective measure of one’s health and was reported on the following scale, from 1 to 5: *very poor, poor, fair, good, very health*.

Then, participants were asked to respond regarding whether they can identify the online vision test using a traditional acuity test for font size with ten cards (*yes* = 1, *no* = 0), with the final composite score ranging from 0 to 10. The sensitivity of hearing test was assessed by a trained interviewer, using a voice at a certain volume to read materials and asking respondents whether they could hear it; *no* = 1, *somewhat* = 2, *very clear* = 3. Physical disability was assessed by asking a single question “Do you need psychical aids when going out?” (*almost every time* = 1, *need some help* = 2, *not at all* = 3). *Cognitive functioning* was assessed with 12 items, including two items, i.e., word immediate and delayed recall tests of memory, a serial five-item subtraction test of working memory, and counting backwards to assess attention and processing speed. Composite scores using all the items created a measure of cognitive functioning, ranging from 0 to 12, with higher scores representing better cognitive functioning.

*Social media perceptions* were measured by different aspects of users’ psychological states regarding WeChat, all sub-scales based on previous similar research of Internet adoption in China [36], Internet adoption among older adults, and social media adoption and usage of younger adults [37]. The first perception is the perceived popularity of WeChat (PP), with three items used to estimate the popularity of social media use among family members, relatives, friends and acquaintances, and the general population; for example, “In your estimate, about how many of your relatives, friends, and acquaintances are using the WeChat?” was used. All items used a 5-point scale ranging from 1 (almost none) to 5 (almost everyone), and the average scores were calculated for the analysis. The consistency and reliability of the scale are acceptable (Cronbach α = 0.748).

Then, the perceived characteristics of WeChat (PC) were used with eight items. Four sub-dimensions were measured: *Perceived ease of use* (PEU), for example, “WeChat is very easy to learn and use”; *P**erceived enjoyment* (PE), for example, “I found WeChat is very interesting”; *Perceived usefulness* (PU), for example, “*WeChat is useful to me”*; and *Rela**tive advantage* (RA), for example, “WeChat brings a lot benefits in my life*”.* The items used a 5-point Likert scale ranging from 1 (strongly disagree) to 5 (strongly agree). The average scores were calculated for the analysis. The consistency and reliability of the scale are acceptable (Cronbach α = 0.731).

Finally, *Perceived needs* (PN) of WeChat included 11 items to measure how social and psychological needs drive individuals to make use of social media to obtain gratification. All questions stem from “How much do you feel that WeChat has satisfied your following needs?” These needs included information needs, for example, “to get domestic and international news” and “to obtain personal related information”; networking needs, such as “to connect with others”; and entertainment and expression needs, such as “to get entertainment” and “recording my life”. All items employed a 5-point Likert scale ranging from 1 (completely dissatisfied) to 5 (completely satisfied). The average scores were also calculated for the final analysis. The scale has good consistency and reliability (Cronbach α = 0.893).

### 2.3. Analysis Procedure

We used descriptive statistics and bivariate analysis to characterize the social media access and Internet access patterns. The hierarchical binary logistic regression was then used to identify factors associated with social media access among older adults. The hierarchical linear regression analysis was performed in the three generation groups (older, middle-aged, and younger) to identify factors associated with the diversity and intensity of social media activity engagement. The regression models included case weights to account for differential probabilities of selection, and the case weights took into account the joint distribution of age, gender, and region of the older adults’ population according to the Chinese Population Statistics Year Book (2017). In order to test for a high correlation among independent variables, multicollinearity detection and variance inflation index (VIF) statistical tests were conducted. Most VIF values were smaller than 2.0, and only the VIF value for perceived usefulness was greater than 2.0 and less than 2.2, which suggests that multicollinearity was not a problem in the models. All the data analysis procedures were conducted in SPSS version 22.0.

## 3. Results

### 3.1. Intergenerational Divide in Digital Access

In order to answer the first research question about whether a grey digital divide exists, the intergenerational analysis of the full sample was conducted. Table 1 presents an overview of the intergenerational divide for digital access with regard to Internet and social media. The descriptive analysis of the Internet and social media (WeChat) adoption rate is detailed within each generational cohort. Results in Table 1 reveal a significant generational divide in terms of Internet access. More than 95% of adolescents and younger adults and middle-aged adults have regular Internet access. Among these generations, the social media access enormously overlaps with Internet access, reaching 97.9% and 94.3% respectively. Figure 1 shows that the digital divide between younger and middle-aged generations in relation to access social media was closing in 2017 (*p* < 0.01), whereas the older adults demonstrated a large gap compared with younger and middle-aged generations.

The results showed that there are significant differences between three generations according to the chi-square test presented in Table 1 (*Value* = 1430.830, *df* = 16, *p* < 0.001). However, the gap between the younger and middle-aged generations is narrowing in terms of the basic level of digital access, according to the descriptive results in Table 1 and Figure 1. Despite the fact that the older adults’ digital access has increased dramatically over time, the digital divide remains; as compared to the 95%+ Internet/social media adoption rate among the younger and middle-aged generation, only 80% of the young older adults (55–59) have Internet/social media access, and the proportion drops to 50% in the older group (60–69) and 30% in the oldest group (70+).

### 3.2. Explaining Older Adults’ Social Media Adoption

The divide for both Internet access and social media access revealed that the access of older adults’ is still an issue and needs to be addressed. In order to further examine factors associated with older adults’ access behavior regarding social media (688 users, 709 non-users), binary logistic regression analysis was employed to investigate social-demographic factors and physical-mental factors relating to WeChat adoption. The social media perception factors were not included in the analysis since the non-users had no further responses to the questions in the surveys of perceptions of social media. The results are presented in Table 2.

In Model 1, only the demographic characteristic factors were included to explain social media access of the older generation. Although Model 1 correctly predicted 73% of the classification, the Homers and Lemeshow Test showed the data were not fitted to the model very well (*p* < 0.05). Therefore, we moved forward to the second step by adding social-economic status variables (education level, living region, and professional rank). After controlling for the demographic variables, Model 2 had a good fit according to the Homers and Lemeshow test (*p* > 0.10), and the variance explained by the model increased by 10 percent.

Finally, Model 3 had a good Homers and Lemeshow fit (*p* > 0.25). After controlling for the demographic and social economic status variables, more information on older adults’ health conditions was included. Findings suggested that there is an independent effect of the general self-rated physical health and cognitive functioning on the likelihood of being a social media user. A one unit change in the predicators of self-reported health and cognitive functioning increased the odds of being a social media user by a factor of around 1.27 and 1.08, respectively, indicating that, the better the older adults’ physical health condition and mental condition, the more they are likely (by 1.27 and 1.08 times) to be a social media user.

Regarding the controlled demographic variables, age and gender were negatively associated with social media adoption; being older increased the odds of being a non-user by a factor of 0.87, while being a male increased the odds of being a non-user by a factor of 0.67. On the contrary, being an urban resident increased the odds of being a user by a factor of 0.32; in addition, the closer the neighborhood in which people lived to a city center, the more likely older adults (0.89 times) were to be a social media user. Furthermore, those who had more children were more likely to adopt social media; a one unit increase in the number of children increased the odds of being a user by a factor of around 1.26.

In terms of social economic status variables, the education level and living region’s economic status were found to be two significant predicators for adopting social media. The results indicated that older adults with a higher education level tended to report using social media (*OR* = 2.03), and generally those who lived in the more developed regions were more likely to be a user (*OR* = 0.71).

### 3.3. Intergenerational Divide in the Diversity of Social Media Engagement

Concerning the third question in relation to diversity of social media engagement, we conducted a two-step analysis. Preliminarily, one-way ANOVA (*F* = 494.866, *df* = 2, *p* < 0.001) was used to discern the difference between generations in terms of the outcome variable of diversity scores to social media activities. A significant generational difference (*p* < 0.001) was found between the younger generation (*M* = 16.82, *SD* = 2.89), middle-aged generation (*M* = 15.25, *SD* = 3.0), and older generation (*M* = 11.68, *SD* = 4.47). Second, hierarchical linear regression analysis was used to test the intergenerational difference in the predictors associated with diversity (only users were included in the analysis).

Table 3 shows the results of hierarchical linear regression analysis of the diversity of social media activity engagement across three generations. The model finally explained 40.1% and 26.0% of the total variation for older and middle-aged people, respectively, whereas Model 3 only explained 5.4% of the variance for the younger generation. For older adults, the controlled variables in Models 1 and 2 accounted for a large variance of 26.4%, indicating that social-demographic variables are key factors in determining older adults’ social media engagement in terms of diversity of activities. Among these significant variables, age (*β* = −0.27, *p* < 0.001) was negatively related to the outcome variable of diversity, whereas residency (*β* = 0.08, *p* < 0.05), education level (*β* = 0.18, *p* < 0.001), and professional rank (*β* = 0.12, *p* < 0.01) were found to be positively associated with the diversity of WeChat activities. This suggests that older adults with an urban residency, who are younger and have a better education and a professional background, were more likely to engage in diverse social media activities.

Furthermore, the social media perception factors contributed about 14% of the variance of the final model, which means these factors are also important determinants of the outcome variable for older adults. Of these, the perceived needs (*β* = 0.25, *p* < 0.001), perceived ease of use (*β* = 0.16, *p* < 0.001), and relative advantage (*β* = 0.07, *p* < 0.05), ranked from large to small, were positively associated with the diversity of engaging in social media activities.

Similarly, for middle-aged generations’ diversity of social media activities, the controlled social-demographic factors accounted for 18.8% of the total variance. The educational level (*β* = 0.22, *p* < 0.01), living region (*β* = 0.08, *p* < 0.05), and age (*β* = −0.02, *p* < 0.01) stand out as the key predicting factors. This indicates that middle-aged people having better educational backgrounds, living in more developed regions, and being younger were more likely to engage in more diverse WeChat activities. In addition, three social media perception factors—perceived ease of use (*β* = 0.20, *p* < 0.001), perceived needs (*β* = 0.19, *p* < 0.001), and perceived popularity (*β* = 0.10, *p* < 0.05)—were found to significantly predict the diversity of WeChat activities. In total, these factors contributed 7.2% of the variance of the predicated variable. The results suggest that the more the middle-aged generation perceived that WeChat is easy, popular, and necessary, the greater the diversity of the activities engaged in by those people on WeChat.

For the younger generation, Model 3 seemed to have less predicting power (a total of 5.4% of variance explained). Only three factors—education level (*β* = 0.17, *p* < 0.01), perceived ease of use (*β* = 0.06, *p* < 0.05), and residency (*β* = 0.06, *p* < 0.05)—were found to significantly predict younger adults’ diversity of WeChat activities. This finding suggests that the younger people having a higher level of education and urban residency, and who perceived WeChat to be easy to use, tended to engage in more diverse social media activities.

### 3.4. Intergenerational Divide in the Intensity of Social Media Engagement

In order to answer the fourth question in relation to intensity, one-way ANOWA analysis (*F* = 52.822, *df* = 2, *p* < 0.001) was first conducted, in which a significant generational difference regarding the outcome variable of intensity score was revealed. The younger generation (*M* = 3.15, *SD* = 0.87), middle-aged generation (*M* = 2.92, *SD* = 0.84), and older generation (*M* = 2.73, *SD* = 0.86), from high to low scores, showed a significant intergenerational difference (*p* < 0.001). Then, hierarchical linear regression was employed to examine the determinants of the three generations’ intensity of WeChat use. Intergenerational differences were also found in the predicators for explaining the intensity of social media engagement. The details shown in Table 4 outline these differences.

Table 4 shows that Model 3 explained a total variance of 35.1%, 20.1%, and 20.5% for older, middle-aged, and younger generations, respectively. For the older adults, the controlled variables of social-demographic factors accounted for a higher variance (*R square* change = 6.2%), whereas these controlled variables contributed a relatively lower variance (3.8% both) for the middle-aged and younger adults’ intensity of social media engagement.

For the older adults, the neighborhood type (*β* = 0.09, *p* < 0.05), number of children (*β* = −0.09, *p* < 0.05), and vision functions (*β* = 0.09, *p* < 0.05) were found to be the significant determinants of the intensity of social media engagement. This indicates that older adults with good vision and fewer children, and living in a neighborhood close to a city center, were more likely to be heavy users of WeChat. In contrast, only one social-demographic variable, gender (*β* = 0.09, *p* < 0.05), was significantly correlated with the intensity of the middle-aged group’s use of WeChat. Furthermore, education level (*β* = 0.12, *p* < 0.001), gender (*β* = 0.10, *p* < 0.001), and residency (*β* = −0.06, *p* < 0.05) were found to be significant predictors of intensity of WeChat engagement.

As shown in Model 3, social media perception factors were found to be the most outstanding determinants of intensity in WeChat engagement after controlling for the social-demographic variables. These factors contributed a large proportion of the variance for each generational group’s intensity of social media engagement. The R square changed by around 29% for the older generation, compared to 16.3% (middle-aged generation) and 16.7% (younger generation), by introducing perceived social media characteristic factors.

The results revealed that perceived needs (*β* = 0.30, *p* < 0.001), perceived enjoyment (*β* = 0.18, *p* < 0.001), and relative advantage (*β* = 0.16, *p* < 0.001) are the most significant determinants of intensity in WeChat use for older adults. This suggests that older adults who perceived that WeChat was needed, easy, and enjoyable were more likely to be intensive users of social media. Furthermore, perceived needs and enjoyment for both middle-aged and younger groups were the significant determinants of intensity of WeChat use. These results indicate that the greater the perceived need and enjoyment of WeChat of both middle-aged and younger individuals, the more likely it is that these two groups will intensively use WeChat.

## 4. Discussion

In recent years, social media have become pervasive throughout society. In particular, WeChat has gained significant popularity because it provides complete daily services for all ages based on the rapid mobile Internet development in China [9]. However, the digital divide exists and even evolves when older people face the new (social) media challenges, in addition to the opportunities [8,11].

Drawing upon the national research surveys across three generational cohorts, our study contributed to the scholarly agenda of the digital divide in many ways. First, the study revisited the digital divide in social media access and engagement, going beyond the most studied first-level digital divide. The prior Internet access approach neglected individuals’ actual online activities; hence, it has been criticized by many scholars [14].

Secondly, in addition to exploring the second level of the digital divide in social media access [6,14], the present study further examined the third digital divide, namely, that in social media engagement. The study operationalized social media engagement into two measurable dimensions: diversity of activity and intensity of engagement. Exploring the diverse and deeper engagement with social media is particularly relevant in the context of the rapidly evolving new media environment, because of the emerging scholarly discussion regarding the benefits of social media for seniors’ later life, which have gained enormous attention globally [8,11].

Thirdly, previous studies have only focused on one generation group in the field of digital divide research [14,38], whereas we provided a more nuanced look at the digital divide related to generational differences. This approach also sheds light on the variance across a full range of the population from a Chinese perspective.

Finally, the present study renewed the research into the digital divide in China by implementing large-scale surveys in mainland China. Since a relatively large number of studies on social media have been conducted using Western samples and little is known in the China context, we provided a Chinese sample to better understand social media access and use [39].

The main results showed a clear picture of a digital divide related to aging. Specifically, the gap between the older and the younger generation, and the gap between the older and middle-aged generations, remain significant. In contrast, the digital gap between younger and middle-aged people seems to be closing.

The grey digital divide stands out and requires further explanation. We found that social-demographic factors, including age, gender, rural/urban residency, neighborhood distance, education level, and living region economic development, significantly predicted seniors’ social media access. Most findings (except for the reverse gender variance) are consistent with previous national research on the digital divide with regard to Internet and social media access [3,5,40]. In contrast to the classic conclusion in diffusion of innovations research that men are more active than women in adopting new media [41], this research found women surpassed men in adopting WeChat. A possible explanation is that older ladies are more active in their social lives after retirement, and are thus are more likely to know and try new things. Moreover, the particular issue of health was examined in the study, and the results implied that older adults’ general health problems and impairment of cognitive function may hinder their social media adoption. This finding is also in line with prior studies on the digital divide among seniors [3,6].

Further examination of the third digital divide via comparison of the diversity of using social media across different generations revealed interesting findings. In accordance with the prior research [14], the social-demographic variables demonstrated a main significant effect on the diverse use of both older and middle-aged adults: the social media activities of those having higher social-economic status and younger age, and living in more developed regions, was more diverse. Again, the gender impact was reversed here: older and middle-aged females surpassed males in using diversified functions of WeChat. However, there was no gender difference in the younger generation. Actually, the existing social-demographic variations (except regarding residency and education level) showed little effect on the diversity of the younger generation’s social media engagement. A possible explanation is that the younger generation is quite digitally fluent due to their never-ending exposure to digital technologies. As a result, their digital skills are better and they can process information as “multi-taskers”, in contrast to previous generations [42]. Furthermore, these background factors and technology barriers are no longer important, and the younger generation tends to be more digitally homogeneous, both in the background and in their social media behaviors [13,43].

In comparison with social-demographic factors, social media perception factors contributed most of the variance in intense use for all generational groups. Those who perceived social media as being necessary and an enjoyable experience in daily life were more likely to become heavy users. This finding also aligns with the previous literature on technology diffusion and social media adoption across different age groups [31]. However, perceived relative advantage was significant only for the older generation, whereas education level was significant only for the younger generation. In particular, older adults’ vision impairment may hinder the intense engagement of social media. It seems that different generations are influenced by different determinants of social media intensity.

## 5. Conclusions

In conclusion, the empirical findings outlined that the digital divide is aging from a cross-generational perspective. This study enriched our understanding of the relationships between demographic, social, economic, and/or media perceptions, and the digital access and social media engagement, in a multidimensional way. First, our study revealed that digital access, both to the Internet and social media, remains a major issue for the aging population in China. Older adults with disadvantaged social-economic status and poorer physical and mental health are less likely to have digital access, and more likely to disengage from the more diverse activities in social media. This grey digital divide raises the concern that digital exclusion may further exacerbate social inequality [6]. Second, once digital seniors are online, the perceived social media characteristics and needs may greatly motive them to use social media more intensely. Third, the generational differences in middle-aged and younger people deserve serious consideration. This divide may contribute to different life stages with complex sociocultural and social-economic interactions.

A digital media-saturated environment may be necessary to meet the demands of the rapidly aging society, not only in China but globally, since digital technologies provide various digital solutions to support older adults’ later lives [2]. As a technology giant, WeChat has merged all daily services for all age groups, which provides assistance to older adults in independent living, and helps them develop social ties with a variety of connections [31]. Social media connects senior citizens’ daily life with social participation and social inclusion [32]. Therefore, the policy makers and technology developers for older adults would benefit from these results.

It is important for policy makers to note that the existing social-economic inequities, such as the rural–urban gulf and the east–west regional divide, may deepen the aging-based digital divide, and eventually lead to severe social exclusion for the aging population. Moreover, it is helpful for technology developers and designers to focus on the characteristics and needs of digital solutions [2,11]. For instance, the design of digital media for older adults should meet the needs of their daily life, and be easy and enjoyable to use. Overall, in order to build an active aging society, including seniors in this ever-changing world of digitalization and removing the social economic and technological barriers are the crucial issues. To achieve these goals, it is necessary to take advantage of social media for older adults to close the aging-based digital gap.

## Figures and Tables

**Figure 1 ijerph-19-12965-f001:**
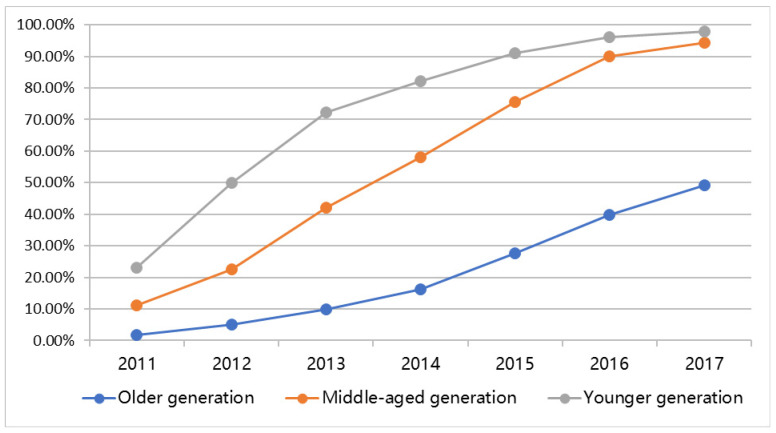
The social media (WeChat) adoption (cumulative %) among three generations over time; results from the present research surveys.

**Table 1 ijerph-19-12965-t001:** Digital access rate (percentage) within each generational cohort.

	Full Sample	Internet Access	Social Media Access
**Generational Cohorts**		***	***
Older generation	**n= 1399**	**52.6%**	**49.1%**
80+		29.4%	15.1%
70–79		31.4%	30.2%
60–69		55.6%	51.3%
55–59		82.8%	80.0%
Middle-aged generation	**n = 543**	**95.9%**	**94.3%**
50–54		82.8%	91.0%
40–49		96.9%	96.0%
Younger generation	**n = 1109**	**98.9%**	**97.9%**
30–39		99.1%	99.1%
20–29		99.7%	99.8%
14–19		94.2%	85.8%

Note: Chi-square test for three generational cohorts’ differences for digital access; *** *p* < 0.001.

**Table 2 ijerph-19-12965-t002:** Binary logistic regression of social media access among older adults (n = 1399).

	Model 1			Model 2			Model 3		
	*β*	se	OR	*β*	se	OR	*β*	se	OR
** *Demographic* **									
Age	−0.13 ***	0.01	0.86	−0.14 ***	0.01	0.87	−0.14 ***	0.01	0.87
Gender(female)	−0.06	0.13	1.07	−0.36 *	0.15	0.69	−0.40 **	0.15	0.67
Marital status(single)	0.36	0.30	0.70	0.03	0.32	0.97	0.07	0.33	0.93
Residency (rural)	1.48 ***	0.16	0.23	1.09 ***	0.17	0.34	1.13 ***	0.18	0.32
Number of Children	0.01	0.06	1.01	0.23 **	0.07	1.25	0.23 ***	0.07	1.26
Neighborhood	0.22 ***	0.05	0.80	0.14 *	0.06	0.87	0.12 *	0.06	0.89
** *SES* **									
Education level				0.72 ***	0.07	2.06	0.71 ***	0.07	2.03
Living regions				0.30 **	0.11	0.74	0.35 ***	0.11	0.71
Professional rank				0.03	0.04	1.03	0.03	0.05	1.03
** *Physical health* **									
Self-reported health							0.24 **	0.07	1.27
Vision							0.102	0.07	1.11
Audition							0.08	0.22	0.92
Has disability							0.21	0.27	1.23
** *Mental health* **									
Cognitive functioning							0.08 *	0.03	1.08
Homers and Lemeshow test		** *0.001* **			** *0.108* **			** *0.256* **	
Pseudo *R*^2^		** *0.40* **			** *0.50* **			** *0.51* **	
Correct Classification		** *0.73* **			** *0.77* **			** *0.78* **	

Notes: 709 samples in category 0, 688 samples in category 1; * *p* < 0.05, ** *p* < 0.01, *** *p* < 0.001.

**Table 3 ijerph-19-12965-t003:** Hierarchical linear regression analysis of the diversity of social media engagement.

	Model 1			Model 2			Model 3		
		*β*			*β*			*β*	
	O	M	Y	*O*	M	Y	*O*	M	Y
** *Demographic* **									
Age	−0.35 ***	−0.25 ***	−0.01	−0.34 ***	−0.24 ***	−0.03	−0.27 ***	−0.02 **	−0.02
Gender(female)	−0.02	0.02	0.02	0.06	0.06	0.01	0.05	0.04	0.01
Marital status(single)	0.02	−0.04	−0.10 *	0.002	−0.03	−0.07	0.01	−0.04	−0.07
Residency(rural)	0.26 ***	0.15 ***	0.12 **	0.15 ***	0.01	0.06 *	0.08 *	0.01	0.06 *
Number of Children	−0.05	−0.09 *	-	−0.001	−0.03	-	0.03	−0.004	-
Neighborhood	−0.06	-	-	0.012	-	-	0.01	-	-
** *SES* **									
Education level				0.23 ***	0.28 ***	0.18 ***	0.18 ***	0.22 **	0.17 **
Living region				0.008	0.10 *	0.02	0.03	0.08 *	0.03
Professional rank				0.13 **	0.06	0.03	0.12 **	0.06	0.03
** *Physical health* **									
Self-reported health							0.03		
Vision							−0.01		
Audition							0.02		
Has disability							0.01		
** *Mental health* **									
Cognitive functioning							0.03		
** *Social media perceptions* **									
*Perceived popularity*							0.04	0.10 *	0.02
*Perceived characteristics*									
Perceived ease of use							0.16 ***	0.20 ***	0.06 *
Perceived usefulness							−0.001	0.01	0.01
Perceived enjoyment							−0.004	0.02	−0.01
Relative advantage							0.07 *	0.06	−0.05
*Perceived needs*							0.25 ***	0.19 ***	0.05
R square change	*0.19*	*0.097*	*0.026*	*0.074*	*0.081*	*0.026*	*0.137*	*0.072*	*0.002*
R square	** *0.19* **	** *0.097* **	** *0.026* **	** *0.264* **	** *0.188* **	** *0.052* **	** *0.401* **	** *0.260* **	** *0.054* **

Notes: O = Older generation (n = 709), M = Middle-aged generation (n = 513), Y = Younger generation (n = 1092); The male/ rural/ single or widowed in the parenthesis refers to the reference group; ** p* < 0.05, ** *p* < 0.01, *** *p* < 0.001.

**Table 4 ijerph-19-12965-t004:** Hierarchical linear regression of the intensity of social media engagement.

	Model 1			Model 2			Model 3		
		*β*			*β*			*β*	
	O	M	Y	*O*	M	Y	*O*	M	Y
** *Demographic* **									
Age	−0.06	−0.07	0.06	−0.07	−0.08	0.03	−0.001	−0.01	0.015
Gender(female)	0.02	0.14 **	0.15 **	0.06	0.14 **	0.14 ***	0.01	0.09 *	0.10 ***
Marital status(single)	−0.05	0.06	0.002	−0.05	0.06	0.02	−0.05	0.04	0.03
Residency(rural)	0.13 **	0.05	−0.05	0.09 *	0.06	−0.08 *	0.01	0.03	−0.06 *
Number of Children	−0.12 **	0.02	-	−0.10 *	0.14	-	−0.09 *	0.003	-
Neighborhood	0.07	-	-	0.09 *	-	-	0.09 *	-	-
** *SES* **									
Education level				0.08	0.03	0.12 ***	0.02	0.04	0.12 ***
Living region				0.002	0.01	0.002	0.01	0.01	0.002
Professional rank				0.09 *	0.06	0.04	0.07	0.01	0.04
** *Physical health* **									
Self-reported health							0.03		
Vision							0.09 *		
Audition							0.03		
Has disability							0.002		
** *Mental health* **									
Cognitive functioning							−0.03		
** *Social media perceptions* **									
*Perceived popularity*							0.05	0.03	0.007
*Perceived characteristics*									
Perceived ease of use							0.02	0.07	−0.03
Perceived usefulness							0.05	0.06	0.07
Perceived enjoyment							0.18 ***	0.14 ***	0.15 ***
Relative advantage							0.16 ***	0.07	0.03
*Perceived needs*							0.30 ***	0.35 ***	0.31 ***
R square change	*0.047*	*0.034*	*0.025*	*0.015*	*0.004*	*0.013*	*0.289*	*0.163*	*0.167*
R square	** *0.047* **	** *0.034* **	** *0.025* **	** *0.062* **	** *0.038* **	** *0.038* **	** *0.351* **	** *0.201* **	** *0.205* **

Notes: O = Older generation (n= 709), M = Middle-aged generation (n = 513), Y = Younger generation (n = 1092); The male/rural/single or widowed in the parenthesis refers to the reference group; * *p* < 0.05, ** *p* < 0.01, *** *p* < 0.001.

## Data Availability

Data could be accessed via contacting the authors upon approval.

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
