# Peer review of "The Digital Divide Is Aging: An Intergenerational Investigation of Social Media Engagement in China"

_ijerph, 2022, doi:10.3390/ijerph191912965_

Round 1

Reviewer 1 Report

article

The Digital Divide is Aging: An Intergenerational Investigation of Social Media Engagement in China

Yuqiong Zhou, Tao He2 and Feng Lin

In general terms, the article is correct, it is well presented and well structured, so its publication is proposed to the editor, provided that some of the minor considerations listed below are met:

MINOR CONSIDERATIONS ON THE TEXT

Two minor errors have been noted in the text:

- In the comment on Table 3, page 9, write “Mode”, although I think you should put “Model”

- In Tables 3 and 4 write “perceived usefulness”, although I think you should put “Perceived usefulness”

CONSIDERATIONS ON THE CONTENT OF SOME TABLES AND THE TEXT

Some values ​​contained in some tables have been observed that we believe do not coincide with what is referred to in the text, so they can cause confusion in their reading. Specific:

- In Table 2, block referring to Model 3, row Education level, the value 2.03 is referred to, when in the text it seems that this value is 2.01 (page 9).

- In Table 3, Model 3, β =-.27*** appears, while β =-2.7 appears in the text.

- Again in Table 3, the level of significance p associated with the values ​​of β in group M for Education level and Living region does not coincide with what is reported in the text (see page 10)

- In the same location as the previous paragraph, for Perceived popularity the same thing happens. We must make a similar comment for Education level (group Y).

The previous observations and other similar ones that could be found, make it recommended that the referred tables should be reviewed and corrected or the comments adjusted to what is contained in the tables.

Reviewer 2 Report

This paper explores the digital divide in social media engagement among the three generations. The manuscript is well written and the topic is worth examining. I have a few points the authors may wish to consider.

1.     WeChat has more multiple functions than other apps like Facebook. However, the functions are not clear for non-users. The diversity scale somewhat helps readers know the functions, but this reviewer thinks making a list of functions of WeChat is helpful to grasp what people can via the app.

2.     There are some scales consisting of several items such as diversity and intensity scale. All the information (e.g. items) should be described in the manuscript or as supplementary files. The same goes for other variables.

3.     Homers and Lemeshow fit test is used but is not described in Analysis procedure.

4.     The hierarchical linear regression was performed in the three generation groups. This should be clearly described in the Method section because the current one is somewhat confusing.

5.     Table 3 shows coefficients of linear regression analysis where diversity scale is the outcome. The scale ranges from 0 – 18, but there is no information about the scale in this study population such as the mean, sd, or distribution. This makes it difficult to interpret the coefficients: are they big or small? The same problem exists in Table 4. These are due to the lack of descriptive data of variables.

6.     Discussion is somewhat redundant because a large part of it refers to the findings of this study which have already described in the Results section.

Reviewer 3 Report

The aim of this study is to answer three relevant research questions: this study addressed four important research questions as follows: is there a digital divide in terms of social media access across three generational cohorts?; what factors explain older adult’s social media access if the grey digital divide exists?; what factors associated with the diversity of social media engagement and how these factors vary by generations?; and what factors associated with the intensity of social media engagement and how these factors vary by generations? All of these research problems are current and relevant.

I have no negative comments on the theoretical justification of the study or the reiew of the literature. I think they are fine.

Regarding the method section:

·        The sample is representative of China’s population. However, as households are sampled and not persons, data are not, in general, independent. Rather, these data are clustered. This is true when generations are compared even more.

·        The differences in sample sizes between the three ages considered are surprising. May the authors elaborate on this?

·        The authors claim that “Social Media Engagement was assessed by a variety of observed variables and latent variables”. Their description s not adequate. Only observed variables are presented, not the latent variables they are going to use. This should be completely specified.

·        Measures lack consistency and reliability estimates.

Results section

·        When a test statistic is presented, we need to see value, degrees of freedom and effect size. This is not the case, for example with the chi-square test presented in table 1.

·        Regarding the results of the binary logistic regressions (models 1 to 3, which I guess are one hierarchical logistic regression) there are several points. First, we need to know how many people are in category 0 and how many in category 1. Otherwise, the correct classification data are meaningless. Second, which Pseudo-R is offered and why?. Third, the differences in fit between model 2 and 3 are negligible and therefore, model 2 is the one to retain.

·        I do not know where this data comes from: the authors say “The results indicated that older adults with higher education level tend to report have social media use (OR= 2.01)”. However, I see in Table 2 coefficients of 2.06 and 2.03.

·        In Table 2, exp(b) should be named OR to keep same name as in the text.

·        In the hierarchical regression differences in R-squares among models in the hierarchy should be tested and presented.

Minor points

Please, try to avoid elders or elderly through the manuscript

In sum, the manuscript may be worth publishing, but the authors must answer my concerns and take the appropriate steps to make the changes.
